# Evaluation of Two Commonly Used Field Tests to Assess *Varroa destructor* Infestation on Honey Bee (*Apis mellifera*) Colonies

Marco Pietropaoli [1,*], Ivana Tlak Gajger [2], Cecilia Costa [3], Dariusz Gerula [4], Jerzy Wilde [5], Noureddine Adjlane [6], Patricia Aldea Sánchez [7], Maja Ivana Smodiš Škerl [8], Jernej Bubnič [8] and Giovanni Formato [1]

1   Istituto Zooprofilattico Sperimentale del Lazio e della Toscana "M. Aleandri", Via Appia Nuova 1411, 00178 Rome, Italy; giovanni.formato@izslt.it
2   Laboratory for Honeybee Diseases–APISlab, Department for Biology and Pathology of Fish and Bees, Faculty of Veterinary Medicine University of Zagreb, Heinzelova 55, 10 000 Zagreb, Croatia; ivana.tlak@vef.hr
3   CREA Research Centre for Agriculture and Environment, Via di Saliceto 80, 40128 Bologna, Italy; cecilia.costa@crea.gov.it
4   Research Institute of Horticulture, Apiculture Division, Kazimierska 2A, 24-100 Puławy, Poland; dariusz.gerula@inhort.pl
5   Department of Poultry Science and Apiculture, Faculty of Animal Bioengineering, University of Warmia and Mazury in Olsztyn, Słoneczna 48, 10-957 Olsztyn, Poland; jurwild@uwm.edu.pl
6   Department of Agronomy, Faculty of Science of Nature and Life, M'Hamed Bougara University, 35000 Boumerdes, Algeria; adjlanenoureddine@hotmail.com
7   Centro de Estudios Apícola (CEAPIMAYOR), Facultad de Ciencias, Universidad Mayor, Camino La Pirámide, Santiago 5750, Chile; patricia.aldea@gmail.com
8   Agricultural Institute of Slovenia, Hacquetova ulica 17, 1000 Ljubljana, Slovenia; Maja.Smodis.Skerl@kis.si (M.I.S.Š.); jernej.bubnic@kis.si (J.B.)
*   Correspondence: marco.pietropaoli@izslt.it; Tel.: +39-06-79099-328

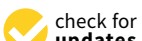



**Featured Application: Identify methods to assess *Varroa* infestation that could be suggested to beekeepers as a suitable IPM tool for varroa control or adopted for selection/research purposes.**

**Abstract:** Assessment of colony infestation by *Varroa destructor* is a crucial part of the Integrated Pest Management (IPM) applied to beekeeping. Natural mite fall, quantified by counting the mites on sticky sheets, is considered a reference method to estimate varroa infestation level in honey bee colonies. However, in recent years, alternative methods that can be used in field conditions have been investigated. In this paper, we report the results of the evaluation of two different methods to estimate the levels of varroa infestation. The experiment involved 151 honey bee colonies in nine apiaries of four countries (Algeria, Croatia, Italy, Poland). After the main honey flow, we compared the 10-day natural mite fall and the powdered sugar roll methods with the varroa population in each colony. According to our results, the powdered sugar roll method could be suggested to beekeepers as a suitable IPM tool for varroa control, while natural mite fall represents a more accurate method that could be adopted for selection/research purposes.

**Keywords:** *Varroa destructor*; powdered sugar roll; natural mite fall; IPM

## 1. Introduction

Estimation of *Varroa destructor* mite infestation rate is an important tool for apicultural practices and studies. It can be used to determine the appropriate timing for treatments, to assess the efficacy of the adopted varroa control strategies and, last but not least, to assess the genetic potential of a colony in terms of varroa resistance. Several methods for varroa mite infestation assessment have been described in detail by Dietemann [1]. A short review of methods adopted for the mite infestation level assessment is reported in Table 1.

**Table 1.** Review of methods for varroa infestation level assessment.

| Method | Reliability | Sample Size | Sensitivity | References |
|---|---|---|---|---|
| Soapy wash | 90% when bees are frozen and centrifuged at 6342 rpm | Approx. 100 bees | More than 3 mites per 100 bees, when the infestation level is lower than 3%, efficiency is 85% | [2] |
| | Equally accurate as powdered sugar test (93%) | 250 frozen bees | | [3] |
| | Positive correlations between relative number of infested honey bees detected by powdered sugar shake and washing bees with alcohol plus soapy water (*r* = 0.90 and *r* = 0.62) | 300–400 bees | | [4] |
| | Hand shaking for 1 min 92% | 300 adult bees | | [5] [1] |
| | Mechanical shaking for 30 min 100% | Approx. 250 adult bees | | [6] |
| Brood uncapping | More reliable in combination with sampling adult bees | 50 worker brood cells and on average 29 drone brood cells | | [7] |
| Powdered sugar roll | 91% | Dusting and $CO_2$ anaesthesia, 761 adult bees (5 replicates) | 10 mites/100 bees, 42 h | [8] |
| | 73.8%/90.98%-low infestation level 76.2%/87.86%-medium infestation level 79.8%/82.16%-high infestation level | Approx. 318 adult bees | Sensitivity in lower infestation levels 84.85%, Sensitivity in medium and high infestation level 100% | [9,10] |
| | 66.10 ± 35.23% and 94.64 ± 9.56% in August and October | 300–400 bees | | [4] |
| $CO_2$ | 49.5% | 200–600 adult bees on average 415 adult bees | At 22 samples from 32 efficiency was below 80% | [11] |
| | 62.5% | On average 427 adult bees | Range 28.6–85.7% | [12] |
| Natural mite fall | Strong linear correlation between natural mite fall and mite fall after chemical treatment r = 0.951 | 22 colonies | | [7] |
| | Strong linear correlation between natural mite fall and mite fall after chemical treatment r = 0.41–0.89 | 150 colonies | Correlation depending on the year | [13] |
| | 76.43% low infestation level 68.26% medium infestation level 66.83% high infestation level | Approx. 318 adult bees | | [10] |

[1] 300 bees are suggested as an optimal sample size for the beekeepers to assess infestation level in their colonies.

The optimal sample size in terms of adult bees and number of hives to sample to estimate the varroa infestation levels at the apiary level, has been determined by Lee et al. [5]. They recommend a sampling procedure in which at least 300 adult honey bees per colony are collected from any comb in the uppermost brood box. Moreover, if a higher accuracy of 0.5 mites per 100 bees is requested, three units of 300 adult bees from each colony, from three separate combs in the upper brood box, should be collected and used to calculate the colony infestation [5].

The powdered sugar roll method was described for the first time by Macedo [10]: 30 g of adult honey bees ($\approx$300 workers) are collected from the open brood area and placed in a jar with a 2 mm hardware cloth or mesh. One tablespoon of powdered sugar is poured through the mesh or cloth, and the jar is rolled for a minute to cover all the adult bees with sugar. Then the jar is turned upside down and shaken for one min so that mites dislodge from the honey bees' body and fall on a prepared plate, where they can be counted. This method is practical, low cost, non-destructive, and has been shown to be effective [1,2,5,10].

"Washing" bees in warm soapy water or ethanol was first described by Fries [14] as a quantitative diagnostic method. It consists of adding one of the mentioned liquids to cover 300 adult honey bees previously sampled and put in a jar. The solution is shaken for 20 s to dislodge the mites from the bees, and the content of the jar is poured over two sieves with different mesh: the first one with large mesh collects all the bees, the second one with finer mesh, placed underneath, collects the mites. The adult honey bees and mites are additionally flushed with large amounts of warm water. The mites and the bees are counted, and the proportion of infested individuals is determined. This method has a low cost, but it is time-consuming on large-scale beekeeping operations and does not permit the survival of the sampled honey bees. Concerning the liquid used, the efficacy of soapy solution (2 mL/L) or 70% ethanol wash in dislodging the mites has been demonstrated [15].

One of the most used methods for assessing varroa infestation is the natural mite fall, which was found to be a reliable method for estimating varroa population size in a colony [7]. It can be quantified in hives equipped with a removable bottom board. The board must be protected by a mesh to prevent active honey bees from discarding the fallen mites. The mesh size should allow the mites to fall through, and the grid should cover as much of the box area as possible. Ants and other insects should be prevented from accessing the bottom board, since they can remove varroa. A sticky sheet or covering with oil can be useful for this aim. According to Bieńkowska and Konopacka [13], the estimation of infestation is more reliable when daily natural parasitic mite fall is calculated from longer periods and tested later in the beekeeping season. The natural mite fall method is labour intensive, costly and allows calculation of a varroa infestation that is related to a specific timeframe.

Objective of this study was to compare the above-mentioned methods with the dispersal, previously generally known as phoretic [16], and total varroa populations in the colonies evaluated by two different treatment strategies. Moreover, the study aimed at identifying pros and cons of each method in order to provide practical recommendations to beekeepers and researchers.

## 2. Materials and Methods

### 2.1. Location and Timing

The study involved 5 researchers, 9 apiaries and 151 colonies in 4 different countries: Algeria (2 apiaries: NA1 and NA2), Croatia (2 apiaries: ITG1, ITG2), Poland (4 apiaries: JW1, JW2, DG1, DG2) and Italy (1 apiary: CC).

Trials were carried out during the active beekeeping season in 2015. Participants started the trial at the end of the main honey flow during summer season, which, considering the different climates, occurred at different dates depending on the apiary location. In each apiary, a minimum of 15 fully developed colonies (consisting of at least ten combs covered with adult honey bees and eight combs of sealed brood in Dadant-Blatt hives) were selected. All colonies were housed in hive boxes with functional screened bottom boards.

### 2.2. Colony Measurements and Sample Collection

Before taking the adult honey bee samples, the number of adult bees and brood cells of each colony was estimated with the Liebefeld method [17]. Bees were selected from one external honey comb covered with the amount of bees sufficient to fill a 120 mL container. In the field, the bee samples were weighted with a scale (1 g accuracy) to reach a minimum of 40 g of adult honey bees.

### 2.3. Powdered Sugar Roll

The powdered sugar roll method estimation was performed on each sample of adult bees (according to [10] modified), adding 35 g of fresh powdered sugar (approximately 2 tablespoons) into the jar, quickly pouring the bee sample from the 120 mL container into a special jar (Dipl.-Ing. (FH) Harald Wössner, Julius-Leber-Strasse 12, 78652 Deisslingen) (Figure 1), closing the cap, rotating the jar for 60 s in order to cover the adult bees with powdered sugar, leaving the jar with bees in vertical position (cap up) for three min and then vigorously shaking the content of the jar (also with sidewall knock) for at least two min through the screen lid, into a sieve that does not permit the passage of *V. destructor* mites.

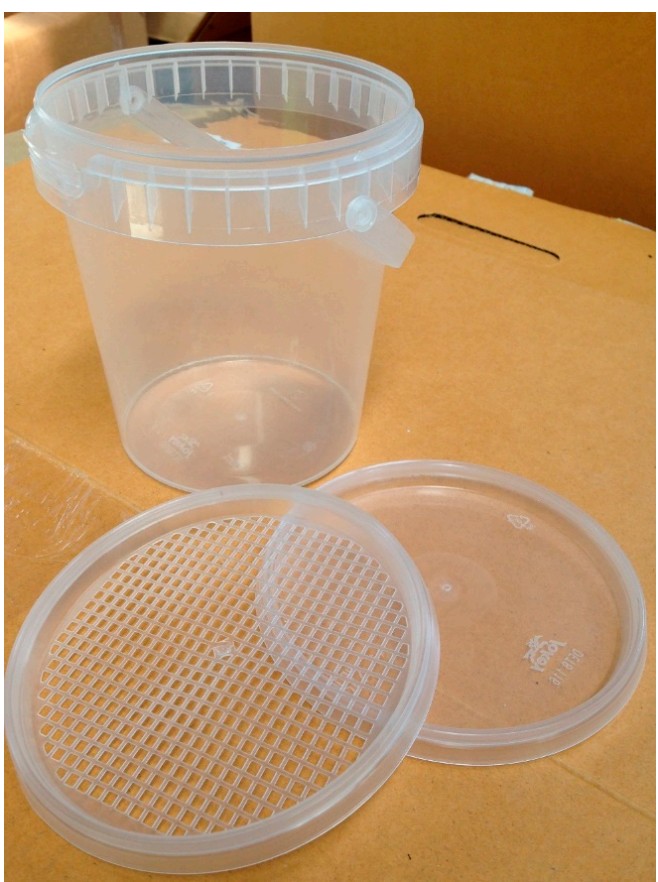

**Figure 1.** Jar provided with a mesh to separate powdered sugar and mites from adult bees.

The main differences from the Macedo [10] procedure were the use of two tablespoons of powdered sugar instead of one, the sampling of 40 g of adult bees instead of 30 g and the three min resting in vertical position of the jar. All those modifications were added in order to increase the number of bees sampled and the probability to dislodge the mites by the powdered sugar.

Instead of being returned to colonies, the samples of shaken adult bees were cooled on ice and taken to the laboratory. The residual number of mites was verified by washing the bees in a soapy solution. The soapy solution was prepared by adding 5 mL of commercial dish liquid soap to one L of water. Each bee sample was shaken into a beaker with 200 mL

of soapy solution. The solution was stirred for 30 min at a speed of 900 rev/min on a magnetic stirrer. The content was subsequently poured through two sieves (one to catch the bees and one the parasitic mites) and a final wash of the bees left in the sieve was performed with high pressure water until no more mites remained on the bees. With the help of a lamp equipped with a magnifying glass, the number of mites in the sieve was counted. The total number of adult honey bees of the samples was counted at the end of the described procedure.

### 2.4. Natural Mite Fall

Natural mite fall was evaluated by counting the mites fallen during 10 days before the powdered sugar sampling. Sticky or oily label sheets were placed on the bottom boards and replaced every second day (or adapted to local conditions). A prerequisite was to take care of preventing access of varroa-eating arthropods in different ways: cutting grass, putting the beehives on specific stands (i.e., legs of stands dipped in jars with water and an oily substance) and by not leaving trays in place during the season but inserting them only at the time of initiation of the trial.

### 2.5. Test Protocols

In order to gather data about the number of mites in dispersal and reproductive phase inside the honey bee colonies and to correlate that information with the infestation level obtained with the samples, two protocols were prepared:

- Protocol 1. Dispersal mite infestation assessment

First, the natural mite fall was assessed for 10 days [1]. On day 10, the honey bee colony strength was estimated [17] and a sample of adult bees from each colony was collected. The samples were analysed with the powdered sugar roll and a subsequent soapy solution wash technique, as described above. The dispersal mite infestation was estimated with the application of an oxalic acid treatment [18] performed on day 10. The oxalic acid solution was prepared using 1 litre of water to dissolve 100 grams of oxalic acid dihydride pure crystals and 1 kg of sucrose and administered to the colonies with a syringe, trickling 5 mL for each intercomb space occupied by bees. The amount of dispersal mites fallen after the treatment was assessed by using sticky or oily label sheets on the bottom boards, replaced every second day (or adapted to local conditions). The counts were performed for five days in a row [1].

A total of 2 researchers (ITG and NA) participated in the trials. Four apiaries in Croatia and Algeria with a total of 60 colonies were involved.

- Protocol 2. Total mite infestation assessment

As for protocol 1, the first step was to assess the natural mite fall for 10 days. On day 10, colony strength was estimated with the Liebefeld method, and a sample of bees from each colony was collected on day 10 and analysed by the powdered sugar roll and soapy solution wash techniques, as described before. The total mite infestation of the colonies was estimated by counting the mite fall on bottom boards from day 10 to day 40. During that period, the queen was caged for 25 days (API.MO.BRU. Campodoro-Italy). Then, at the end of the queen caging period, in absence of brood, the test colonies were treated with oxalic acid (as previously described) [19]. Mite counts on bottom boards were performed until 5 days after the oxalic acid treatment. The sum of the mites fallen during the queen caging period, plus the mites fallen after the treatment, represent the total number of mites in the colony.

A total of three researchers (DG, JW, CC) participated in the trials. Five apiaries with a total of 91 colonies were involved.

Statistical analysis

A one-way ANOVA model was used to test the differences between the apiaries in terms of colony population size (adult bees and brood). For infestation level, Bartlett's test was used for equality variances, and Welch's ANOVA was applied.

The statistical differences between the sampling methods were analysed by Kruskal-Wallis test [20]. To verify how many mites were dislodged with the powdered sugar roll method, Kendall's tau coefficient [21] was used to measure the ordinal association between the number of mites found after the powdered sugar shaking and the number of mites found with the soapy solution washing.

Spearman correlation test [22] was applied to evaluate the methods. Comparisons were made between:

- 10-day natural mite fall and dispersal varroa population (protocol 1) or total varroa population (protocol 2)
- mites found with the powdered sugar roll method and dispersal varroa population (protocol 1) or total varroa population (protocol 2)

With a stepwise regression analysis, it was possible to verify if the factors: number of adult bees, brood, apiary, and tester were significant in the analysis.

Thus, colonies were divided according to the second quartile (median) into two groups in terms of three features: amount of adult bees (weak or strong colonies); amount of brood (low or high); infestation level (low or high infested colonies). Spearman correlation test was applied to evaluate the methods used in both protocols.

Analysis was performed using XLSTAT™ (Addinsoft, Paris, France) and Statistica© ver. 13 (StatSoft, Hamburg, Germany) software.

### 3. Results

The container used in all trials (120 mL) was able to collect $46.3 \pm 8.1$ g of adult bees (Figure 2). Two researchers (CC and DG2) collected statistically heavier samples, compared to the other researchers. The relation between the weights measured with the scales and the actual number of adult bees ranged from 5.9 to 11.0 bees/g (mean $\pm$ s.d. = $8.4 \pm 1.8$).

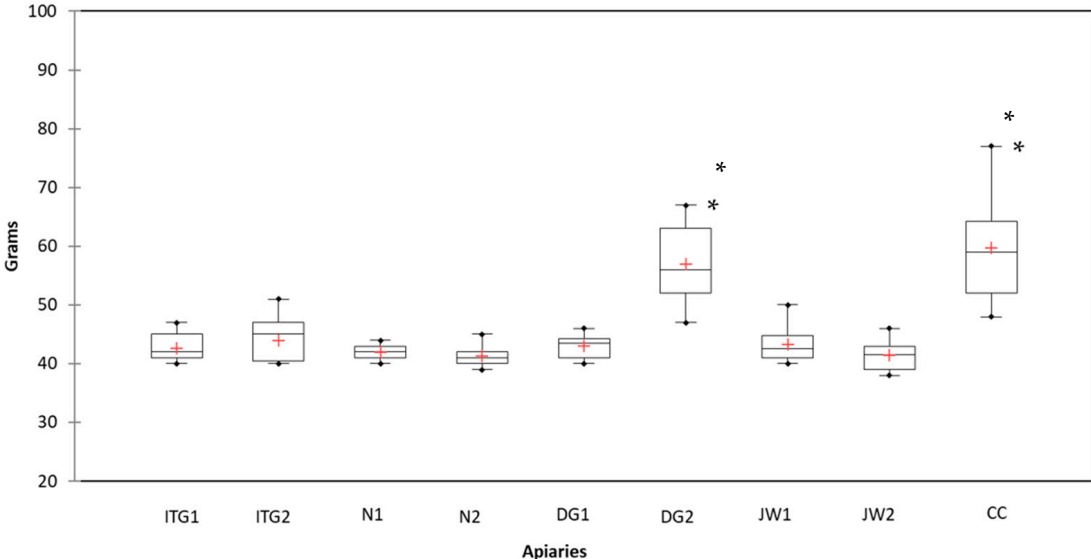

**Figure 2.** Weights of collected samples of adult bees in the different apiaries. * $p < 0.0001$; Bonferroni corrected significance level: 0.0018.

Honey bee colonies in which protocol 1 was applied had a similar ($p = 0.102$) amount of adult bees, but they differed significantly in the amount of brood (Table 2). Dispersal mite infestation was also different and ranged from 26 to 149 mites per colony in apiaries ($p < 0.01$) (Table 2).

**Table 2.** Strength of colonies and dispersal mite infestation in the test apiaries. Correlation between the number dispersal mites in colonies and infestation of samples tested using different methods. Protocol 1

| Apiary ID | Mean Amount of Bees * | Mean Amount of Brood ** | Dispersal Mite Infestation (Mean) *** | Powdered Sugar Roll | | Natural Mite Fall | |
|---|---|---|---|---|---|---|---|
| | | | | Spearman Correlation (*p*-Value) | Coefficient of Determination | Spearman Correlation (*p*-Value) | Coefficient of Determination |
| ITG1 | 16816 a | 24,880 bc | 146 b | 0.564 (0.030) | 0.319 | **0.839 (<0.0001)** | **0.704** |
| ITG2 | 12,055 a | 32,746 c | 26 a | 0.623 (0.015) | 0.388 | 0.345 (0.206) | 0.119 |
| NA1 | 17,076 a | 19,756 ab | 76 ab | −0.146 (0.604) | 0.021 | −0.072 (0.802) | 0.005 |
| NA2 | 13,922 a | 15,293 a | 29 a | −0.462 (0.085) | 0.213 | 0.136 (0.625) | 0.018 |
| Total apiaries: 4 | **14,758** | **23,169** | **69** | **0.368 (0.004)** | **0.136** | **0.537 (<0.0001)** | **0.289** |

\* ANOVA $F_{(3, 56)}$ = 2.16), $p < 0.102$; \*\* ANOVA $F_{(3, 56)}$ = 9.92), $p < 0.01$; \*\*\* Welch's ANOVA $F_{(3, 26.8)}$ = 11,35), $p < 0.01$, Variances in apiaries were different (Bartlett's test for equal variances, $p < 0.01$). Different letters in columns indicate significant differences at $\alpha = 0.05$.

The proportion of mites dislodged by the powdered sugar was very high, as evaluated with by the subsequent soapy solution wash (Kendall Tau on concordance = 0.874). In the apiaries where protocol 1 was applied (dispersal mites infestation predictor), the number of mites collected with the powdered sugar roll method was weakly correlated with the number of dispersal mites counted after the oxalic acid treatment (Table 2). Natural mite fall count was statistically related to the amount of dispersal mites only in one apiary (ITG1) with a coefficient of determination of 0.704. In the other apiaries, natural mite fall was not correlated with the amount of dispersal mites (Table 2).

Results from apiaries NA1 and NA2 were opposite to what expected (negative correlation), so only Croatian apiaries were included in the analysis (Table 3). The highest correlation rates between the number of dispersal mites and infestation of samples tested using the powdered sugar roll method occurred in apiaries where colonies had more brood. For natural mite fall, the highest correlation rate was found in colonies with a lower amount of brood, although statistically significant correlations were present in all conditions (Table 3).

**Table 3.** Spearman correlation between colonies classified according to strength, amount of brood and infestation class, and dispersal infestation level obtained using different methods. Protocol 1. Only Croatian apiaries were included.

| Colony Category | Powdered Sugar Roll | Natural Mite Fall |
|---|---|---|
| Weak | **0.79 (*p* < 0.05)** | **0.66 (*p* < 0.05)** |
| Strong | 0.60 (*p* < 0.05) | **0.74 (*p* < 0.05)** |
| Low amount of brood | **0.66 (*p* < 0.05)** | **0.86 (*p* < 0.05)** |
| High amount of brood | **0.84 (*p* < 0.05)** | **0.67 (*p* < 0.05)** |
| Low infested | **0.67 (*p* < 0.05)** | **0.73 (*p* < 0.05)** |
| High infested | 0.48 (*p* > 0.05) | **0.68 (*p* < 0.05)** |

As for protocol 2, the number of mites detected with the powdered sugar roll method was statistically correlated with the total number of mites in the honey bee colonies only in one apiary (CC) (Table 4). Natural mite fall count was correlated with the total amount of mites in three apiaries, but the coefficient of determination was high only in two of them. In the other apiaries, natural mite fall was not correlated with the number of total mites in the colonies (Table 3). Strength and infestation level varied between apiaries in colonies managed according to protocol 2 ($p < 0.01$; Table 4). The greatest variation among apiaries was observed in mite infestation level that ranged from 118 to 2478 mites per colony in apiaries (Table 4).

**Table 4.** Strength of colonies and total mite infestation in subsequent apiaries. Correlation between the number mites in colonies and the infestation of samples tested using different methods. Protocol 2.

| Apiary ID | Mean Amount of Bees * | Mean Amount of Brood ** | Total Mite Infestation (Mean) *** | Powdered Sugar Roll | | Natural Mite Fall | |
|---|---|---|---|---|---|---|---|
| | | | | Spearman Correlation (*p*-Value) | Coefficient of Determination | Spearman Correlation (*p*-Value) | Coefficient of Determination |
| DG1 | 19,456 b | 32,733 b | 118 a | 0.267 (0.365) | 0.071 | 0.092 (0.773) | 0.008 |
| DG2 | 18,793 b | 28,320 b | 541 b | 0.354 (0.116) | 0.125 | **0.549 (0.011)** | **0.302** |
| JW1 | 16,697 b | 31,167 b | 131 a | 0.044 (0.879) | 0.002 | **0.846 (<0.001)** | **0.715** |
| JW2 | 18,061 b | 16,911 a | 179 a | 0.221 (0.296) | 0.049 | 0.084 (0.695) | 0.007 |
| CC | **12,843 a** | **26,915 b** | 2478 b | **0.890 (<0.0001)** | **0.791** | **0.926 (<0.0001)** | **0.857** |
| Total apiaries: 5 | **17,210** | **25,981** | **673** | **0.547 (<0.0001)** | **0.299** | **0.625 (<0.0001)** | **0.391** |

* ANOVA $F_{(4, 82)}$ = 7.46), $p < 0.001$; ** ANOVA $F_{(3, 56)}$ = 13.96), $p < 0.01$; *** Welch's ANOVA $F_{(4, 33.9)}$ = 3.15), $p = 0.02$. Variances in apiaries were different (Bartlett's test for equal variances, $p < 0.01$). Different letters in columns indicate significant differences at $\alpha = 0.05$.

According to protocol 2, the estimation of bee infestation is highly dependent on the strength of the colonies and the degree of infestation (Table 5).

**Table 5.** Spearman correlation between colonies classified according to strength, amount of brood and infestation class, and total infestation level obtained using different methods. Protocol 2.

| Colony Category | Powdered Sugar Roll | Natural Mite Fall |
| --- | --- | --- |
| Weak | **0.77 ($p$ <0.05)** | **0.77 ($p < 0.05$)** |
| Strong | 0.32 ($p < 0.05$) | 0.30 ($p < 0.05$) |
| Low amount of brood | 0.58 ($p < 0.05$) | 0.56 ($p < 0.05$) |
| High amount of brood | 0.44 ($p < 0.05$) | **0.67 ($p < 0.05$)** |
| Low infested | 0.44 ($p < 0.05$) | 0.26 ($p > 0.05$) |
| High infested | 0.52 ($p < 0.05$) | **0.68 ($p < 0.05$)** |

Considering data from all apiaries that adopted the two protocols (Tables 2 and 4), the correlation between the number of dispersal or total mites and the two on-field methods were statistically significant ($p < 0.05$), but the relation of features was low. Stepwise regression analysis confirmed that other factors, such as the number of adult bees, brood coverage, hive types, and testers, were insignificant in the analysis.

## 4. Discussion

Results obtained from field and laboratory diagnostic procedures can give information about colony treatments against varroosis in a timely manner. Moreover, verifying the effectiveness of control measures helps beekeepers in choosing the veterinary medical product (VMP) suited to their conditions and management, as well as veterinarians who have the professional prerogative of being able to prescribe authorised veterinary medicines [23,24]. Last but not least, results on varroa infestation levels are an important tool in breeding programs for varroa resistance [25].

Protocols were devised in order to have data that could allow making practical considerations also considering the apiary conditions. Despite all researchers using the same container (Figure 1), variable quantities of adult bees were obtained, probably depending on the nectar flow or the width of the cells of wax foundations or the subspecies of the bees, which could have influenced the weight of the samples (CC and DG2, Figure 2).

The use of powdered sugar to dislodge mites from adult bees was effective, as also confirmed by other researchers [3,4,8,10]. The hard shaking of the jar containing the powdered sugar has been proven in previous studies to be an important factor to improve the accuracy [26]. The use of alcohol solutions is effective in dislodging the mites [6], but sixty seconds of rapid agitation might not be feasible for large-scale beekeeping operations. The use of portable agitators can improve the accuracy and reduce the effort [27] and further improvements of the methodology, such as the immersion for 2 min in 91% isopropyl alcohol without shaking the sample, seem very promising alternatives [28]. Other methods used to evaluate varroa infestation, such as ether wash [29] or $CO_2$ test [11], were not considered in our trial for their lack of practicality, high costs or low efficiency in dislodging the mites.

Concerning the dispersal mite infestation of the honey bee colonies (protocol 1), the number of mites collected with the powdered sugar roll method was not always statistically correlated with it and gave more accurate results in colonies that were weaker or less infested (Table 3). Natural mite mortality was related to the amount of dispersal mites only in one apiary (ITG1, Table 2), which was not treated with acaricides for two consecutive years. Correlation coefficients of natural mite fall were higher in strong colonies when the brood amount is lower and independently on hive infestation (Table 3).

Spearman correlation ($p$-value) and coefficient of determination of the results gained as total mite infestation (protocol 2) was very good for powdered sugar roll method in

one apiary (CC), where infestation was significantly higher compared to other apiaries (Table 4). Natural mite fall count was correlated with the total amount of mites in three apiaries (Table 3), but the coefficient of determination was high only in two of them (JW1, CC). These results support previous findings where natural mite fall is strongly correlated with the total number of mites in the colonies [7].

There are also several factors that affect mite population increase in colonies (brood production, life cycle of worker and drone brood, reproductive potential of *V. destructor* mite, etc.) that must be considered in calculations for infestation rate dynamics [30]. Moreover, the estimation of mite fall in summer can be used as a predictor of the infestation rate in autumn [13] and it must be stressed that in this study, parasitic mite infestation levels were estimated as daily natural fall based on mite counts during 10 days, in active beekeeping season. Three-week monitoring periods give the best assessment of colony infestation rates by natural mite mortality [31].

## 5. Conclusions

From the practical point of view, even if a single sample of approx. 400–450 adult bees is not sufficiently accurate to estimate the single colony infestation, as also found by Lee [5], and the whole apiary should be considered by the beekeeper as a single unit, an overall positive correlation between the number of dispersal or total mites and the two on-field methods was found.

The purpose of collecting data on varroa infestation is crucial. Considering that the powdered sugar roll method represents an alternative for beehives not equipped with wire mesh bottoms and has the advantage of giving an immediate response, in highly infested colonies it will certainly detect the threat for their survival (Table 4) and it will give a good prediction of the total infestation in weak colonies (Table 5). For this reason, this method could be successfully adopted by beekeepers to evaluate the need of a treatment during the season.

Infestation between apiaries can vary widely due to different environmental conditions (e.g., re-infestation, drifting) or bee genetics, all factors affecting varroa and bee population dynamics. Mean infestation level in a whole apiary with a precision of C = 0.25 should be estimated with a sample of 200–400 adults examined from one comb in each of eight or fewer colonies in the apiary (e.g., six from an apiary with 20 colonies, five from an apiary with 10 colonies, or three from an apiary of four colonies) [5].

The natural mite fall method appeared to be more accurate, and it was sufficiently reliable to estimate the dispersal and total varroa population in single colonies, in different conditions. Due to the time-consuming efforts required and the prerequisites that need to be implemented to apply it, its use could be suggested for research/selection and breeding purposes.

Finally, beekeepers will be able to choose their own method according to personal experience, apiary size and costs, and, in order to have comparable data, they should always adopt the same method and collect data year after year in an Integrated Varroa Management vision.

**Author Contributions:** Conceptualization, methodology, investigation, writing—original draft preparation; M.P., I.T.G., C.C., D.G., J.W., N.A., P.A.S., M.I.S.Š., J.B., G.F.; software, validation, formal analysis, data curation; M.P., D.G.; writing—review and editing, G.F., C.C.; visualization, M.P.; supervision, G.F. All authors have read and agreed to the published version of the manuscript.

**Funding:** This research received no external funding.

**Institutional Review Board Statement:** Not applicable.

**Informed Consent Statement:** Not applicable.

**Data Availability Statement:** Data is contained within the article.

**Acknowledgments:** The authors wish to thank the Secretary-General of Apimondia R. Jannoni-Sebastianini and the technician G. Husinec for their kind contributions and comments. Special

acknowledgements go to COLOSS (Honeybee Research Association) for personal meetings and fruitful collaboration of the partners.

**Conflicts of Interest:** The authors declare no conflict of interest.

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
