# Peer review of "Evaluation of Two Commonly Used Field Tests to Assess Varroa destructor Infestation on Honey Bee (Apis mellifera) Colonies"

_applsci, doi:10.3390/app11104458_

Round 1
Reviewer 1 Report
This manuscript presents the results of a comparison of two methods (natural mite fall and powdered sugar roll) allowing to estimate the levels of varroa infestation in honeybees.
Main comments and suggestions:
- Please explain the purpose of the study in more detail and indicate what is new in the study.
- What were the guidelines for selecting individual apiaries for Protocol 1 and 2? Why were both protocols not applied in one apiary?
- In my opinion, a separate chapter "Conclusions" should be added in which the application of the described tests in beekeeping will be explained.
Some other comments:
lines 43,57,66: incorrect citation method
lines 29 and 92: why does the word "Country" start with a capital letter?
In table 1 the references are incorrectly written. Please consider presenting the table horizontally, it will make it more readable.
In my opinion, Figure 1 should be removed as it adds nothing to the article.
Author Response
Comments and Suggestions for Authors
This manuscript presents the results of a comparison of two methods (natural mite fall and powdered sugar roll) allowing to estimate the levels of varroa infestation in honeybees.
Main comments and suggestions:
Please explain the purpose of the study in more detail and indicate what is new in the study.
Dear Reviewer,
Thank you for your comments and suggestions. The aim of the study was to verify if powdered sugar method (according to Macedo, modified) and a 10-days natural mite fall were related to the total and dispersal varroa populations in the colonies, calculated with two different protocols (protocol 1 and protocol 2). In literature, no similar strategies to quantify total and dispersal varroa populations (and compare with Varroa infestation levels and amounts of adult bees and brood) were used. Moreover, the involvement of different researchers from different countries permitted to gather many data and identify pros and cons for practical recommendations to beekeepers and researchers.
What were the guidelines for selecting individual apiaries for Protocol 1 and 2? Why were both protocols not applied in one apiary?
Each researcher that decided to participate to the study could select the protocol to follow, depending on availability of time and resources. The two protocols were devised to obtain two different data and could not be merged into one protocol as, for instance, for the dispersal varroa infestation assessment, the caging could influence the varroa dynamics during the 5 days after treatment (as during the caging period there is an increase of mite fall – Giacomelli et al.).
In my opinion, a separate chapter "Conclusions" should be added in which the application of the described tests in beekeeping will be explained.
A chapter “Conclusions” has been added to the manuscript. Thank you for your suggestion.
Some other comments:
lines 43,57,66: incorrect citation method
lines 29 and 92: why does the word "Country" start with a capital letter?
Errors in lines 43,57,66 and lines 29 and 92 have been corrected in the revised manuscript.
In table 1 the references are incorrectly written. Please consider presenting the table horizontally, it will make it more readable.
In my opinion, Figure 1 should be removed as it adds nothing to the article.
Table 1 has been modified according to your suggestion and Figure 1 has been removed.
Best regards
Reviewer 2 Report
Review of the paper Evaluation of on-field methods to assess Varroa destructor mite infestation on honeybees (Apis mellifera). The study presents assessment of infestation of bee colonies using two protocols provided by the Authors. In my opinion, the title of the work should be slightly changed: Methods to assess Varroa destructor mite infestation on honeybees in field.
The main drawback of the study is that the analyses were performed only in one year. Despite the impressive number of apiaries and countries where the research was carried out, it is interesting that the authors report on only one year of experiments and the data seem incomplete. Perhaps, other data have been presented in other papers by the Authors, who may have divided the data to increase the number of publications.
The tables are prepared very carelessly; there is no explanation of the significance of the differences in the rows or columns. Differences are often moved to the consecutive row, which significantly enlarges the cells in the table.
Major reservations:
Table 1 is too large to be included in the text. Please transfer it to the supplementary materials.
L 88 …go more in deep… I do not think that the methods have indeed deepened the knowledge of testing bee colonies.
L 95 … mid-summer, after the main honey flow… The experiment was carried out in different countries with different climates and nectar rewards, which need to be specified together with the exact term.
L 128. How were the bees cooled?
L 130. The washing liquid in each country may have had a different chemical composition and viscosity, which may have produced the different surface tension and thus the shedding of V.d. mites from the body of the bees.
L 160-161. From which countries?
L 286. …but correlation coefficients were better… What did the Authors mean?
L 289. See the comment above.
L 311. Please provide the Lee equation.
Figure 3. Maybe the bee weight also depended on the type of wax foundation in the colonies, the width of the cells, and the subspecies of the bees (see line 269-270).
The authors did not mention the swarming mood during the disease treatment or possible formation of nucleus colonies from the analyzed colonies. Maybe the V.d. mite has already been diluted in the colonies, hence the different result of the infestation of the bee colonies.
Table 2 and 4 provide data on Mean amount of bees, Mean amount of brood. How were they estimated? there is no information in the Methods.
Table 3 and 5. How were Weak colonies assessed? Again, there is no information in the methodology section.
Author Response
Comments and Suggestions for Authors
Review of the paper Evaluation of on-field methods to assess Varroa destructor mite infestation on honeybees (Apis mellifera). The study presents assessment of infestation of bee colonies using two protocols provided by the Authors. In my opinion, the title of the work should be slightly changed: Methods to assess Varroa destructor mite infestation on honeybees in field.
Dear Reviewer,
Thank you for your comments and suggestions. We changed the title of the manuscript into: “Evaluation of two commonly used field tests to assess Varroa destructor infestation on honey bee (Apis mellifera) colonies”
The main drawback of the study is that the analyses were performed only in one year. Despite the impressive number of apiaries and countries where the research was carried out, it is interesting that the authors report on only one year of experiments and the data seem incomplete. Perhaps, other data have been presented in other papers by the Authors, who may have divided the data to increase the number of publications.
Analyses were performed only in one year and all data are presented. No other data from the same trials are or will be presented in other papers.
The tables are prepared very carelessly; there is no explanation of the significance of the differences in the rows or columns. Differences are often moved to the consecutive row, which significantly enlarges the cells in the table.
Major reservations:
Table 1 is too large to be included in the text. Please transfer it to the supplementary materials.
All tables have been revised. We would like to keep table 1 in the main article if permitted by the Editorial Office as it gives an important review of the available methods and practical considerations for readers.
L 88 …go more in deep… I do not think that the methods have indeed deepened the knowledge of testing bee colonies.
Thank you for your note, the whole part has been revised.
L 95 … mid-summer, after the main honey flow… The experiment was carried out in different countries with different climates and nectar rewards, which need to be specified together with the exact term.
The concept has been specified better in the revised article.
L 128. How were the bees cooled?
A specification has been added (“on ice”)
L 130. The washing liquid in each country may have had a different chemical composition and viscosity, which may have produced the different surface tension and thus the shedding of V.d. mites from the body of the bees.
Yes, we considered this aspect and we tried to use similar commercial washing liquids. In discussion are also cited recent papers about the testing of different washing liquids. From the practical point of view, it should be also considered that it is not possible to suggest beekeepers to use all the same identical chemical product if we are talking about “commercial” dish soaps (as also reported in other papers).
L 160-161. From which countries?
A specification has been added
L 286. …but correlation coefficients were better… What did the Authors mean?
L 289. See the comment above.
L 311. Please provide the Lee equation.
The above mentioned sentences have been changed to improve clarity
Figure 3. Maybe the bee weight also depended on the type of wax foundation in the colonies, the width of the cells, and the subspecies of the bees (see line 269-270).
Thank you for your note, we added your suggestion.
The authors did not mention the swarming mood during the disease treatment or possible formation of nucleus colonies from the analyzed colonies. Maybe the V.d. mite has already been diluted in the colonies, hence the different result of the infestation of the bee colonies.
The different infestation levels of the colonies are surely deriving from their management during the whole season, the genetic diversity of bees, etc. but in our study the aim was to test the different methods in a specific time-frame (“at the end of the main honey flow”) in order to gather useful information for beekeepers from the practical point of view (eg. ability of the methods to identify the need of a treatment)
Table 2 and 4 provide data on Mean amount of bees, Mean amount of brood. How were they estimated? there is no information in the Methods.
The number of adult bees and brood cells of each colony was estimated with the Liebefeld method. It was reported in the previous version (in the revised version, please see lines 105-106)
Table 3 and 5. How were Weak colonies assessed? Again, there is no information in the methodology section.
Colonies were divided according to the second quartile (median) into two groups in terms of three features: amount of adult bees (weak or strong colonies); amount of brood (low or high); infestation level (low or high infested colonies). It was reported in the previous version (in the revised version, please see lines 201-203)
Round 2
Reviewer 1 Report
The manuscript has been significantly improved over the original, especially in terms of readability.
In particular, the additions to the Introduction and the new Conclusions section are useful in clarifying the objectives of the research and how the conclusions can be applied by beekeepers and researchers.
Author Response
Dear Reviewer,
thank you for your kind revisions and suggestions.
Best regards,
Marco